# An Effective Secured Dynamic Network-Aware Multi-Objective Cuckoo Search Optimization for Live VM Migration in Sustainable Data Centers

N. Venkata Subramanian [1] and V. S. Shankar Sriram [2,*]

1    School of Computing, SASTRA Deemed-to-be University, Thanjavur 613401, Tamil Nadu, India
2    Center for Information Super Highway (CISH), School of Computing, SASTRA Deemed-to-be University, Thanjavur 613401, Tamil Nadu, India
*    Correspondence: sriram@it.sastra.edu

**Abstract:** With the increasing use of cloud computing by organizations, cloud data centers are proliferating to meet customers' demands and host various applications using virtual machines installed in physical servers. Through Live Virtual Machine Migration (LVMM) methods, cloud service providers can provide improved computing capabilities for server consolidation maintenance of systems and potential power savings through a reduction in the distribution process to customers. However, Live Virtual Machine Migration has its challenges when choosing the best network path for maximizing the efficiency of resources, reducing consumption, and providing security. Most research has focused on the load balancing of resources and the reduction in energy consumption; however, they could not provide secure and optimal resource utilization. A framework has been created for sustainable data centers that pick the most secure and optimal dynamic network path using an intelligent metaheuristic algorithm, namely, the Network-aware Dynamic multi-objective Cuckoo Search algorithm (NDCS). The developed hybrid movement strategy enhances the search capability by expanding the search space and adopting a combined risk score estimation of each physical machine (PM) as a fitness criterion for ensuring security with rapid convergence compared to the existing strategies. The proposed method was assessed using the Google cluster dataset to ascertain its worthiness. The experimental results show the supremacy of the proposed method over existing methods by ensuring services with a lower total migration time, lower energy consumption, less makespan time, and secure optimum resource utilization.

**Keywords:** network-aware VM migration; live VM migration; security; sustainable data center; VM consolidation; dynamic cuckoo search

## 1. Introduction

In light of the advent of cloud computing, virtualization is one of the most appropriate methods of developing cloud data centers. These large-scale cloud data centers include several physical machines, virtual machines, and storage data center services globally, made by cloud providers [1]. Migration of virtual machines without disconnecting clients improves cloud data center performance. This practice is called Live Virtual Machine Migration (LVMM) [2], which attains specific goals such as resource utilization, fault tolerance, load balancing, and power savings. To benefit from this LVMM, there is a need to handle the challenges that arise during LVMM concerning migration costs.

As cloud data users increasingly rely on cloud services, security and network optimization play an essential role in improving efficiency [3,4]. Over the years, most researchers have taken up this network path-selection problem as a multi-objective scenario [5,6]. Such approaches concentrated on constructing a new topology for cloud data centers by applying heuristic algorithms. In turn, tremendous effort has been made in previous research on minimizing the prominent migration factors such as power, network consumption, and

resource wastage, but with few considering these jointly. Nevertheless, the irony of these approaches lies in selecting the shortest path and applying security that results in wastage of bandwidth and the occurrence of SLA violations [7,8]. Hence, they may work well on the topology building but fail to provide consistent and secured network performance in all live migration scenarios.

Over the past few decades, cloud computing has extensively applied the Cuckoo Search (CS) algorithm to solve complex multi-objective optimization problems [9]. It employs Levy flight distribution to search for the global optimum. Compared with the other optimization algorithms, the Genetic Algorithm (GA) and CS play crucial roles in the VM migration network path/topology selection. The standard Cuckoo Search algorithm has been implemented for network selection and resource allocation multi-objective optimization problems for task arrangement in the topology of the cloud network [9]. In [10], the authors developed a multi-objective-based scheduling algorithm based on conventional Cuckoo Search optimization. In this work, the laying and immigration are defined, allowing them to be used in the scheduling scenario of a directed acyclic graph of the problem. A new algorithm was then developed using Cuckoo Search optimization to increase the profitability of the cloud providers; it aims to reduce cloud costs and minimize time wastage, enhancing the data center's performance. According to the standard principles, a strategy with excellent local search capability tends to have weak global search capability. This is proved when solving complex optimization problems; CS rapidly traps the local optimum when the algorithm uses only a single search strategy. This single search strategy leads to a lack of diversity in the search space. As stated in [10,11], the search techniques using multiple strategies are potentially promising.

Therefore, this paper explores a new Dynamic Cuckoo Search multi-objective optimization algorithm for secured network path selection during live virtual machine migration. Our work has three objectives to help obtain effective results to achieve the best possible solutions: maintenance of optimal resource utilization; selecting the shortest net-work path that exists, with optimal network bandwidth usage; and ensure migration authentication. Cuckoo Search approaches propose an advanced strategy for choosing the shortest network path by combining Levy flights and Gaussian distributions. Furthermore, estimating the risk score factor assures the optimal choice of each intermediate node. Thus, the proposed NDCS algorithm is intended to reduce performance deterioration factors such as total migration time, makespan time, energy consumption, cloud resource utilization, and reduced security risks on the cloud. The main contributions of the proposed study are as follows:

- During LVMM, we present a novel dynamic multi-objective Cuckoo Search method to select the optimal secured network path.
- An innovative search strategy based on dynamic cuckoo Levy flights and a Gaussian distribution is proposed.
- To establish the optimal secured path, the combined risk score factor is evaluated.
- The proposed NDCS outperforms the state-of-the-art techniques in terms of performance metrics such as total migration time, makespan time, energy consumption, resource usage, and ensuring risk scores.

This article is structured as follows: Section 1 introduces the necessity of setting up a secure network path for effective Live VM migrations in sustainable data centers. In contrast, Section 2 discusses the existing works that disclose the significance of the proposed work. Section 3 deliberates the working nature of the classic Cuckoo Search algorithm. Section 4 describes the assumptions made about the system model. In Section 5, we provide a discussion around a Network-aware Dynamic Cuckoo Search algorithm (NDCS) for sustainable data centers. Section 6 deals with the experimental setup and presents different results that integrate the viability of the proposed method and the dataset used for evaluation. Finally, Section 7 concludes the article and explores possible future works.

## 2. Related Works

VMs are often relocated from one physical node to another to deliver a service to cloud customers on-demand without disconnecting clients or applications, known as live migration. The purpose of live migrations is to minimize energy consumption (EC), increase resource usage, and decrease the number of migrations and violations of service-level agreements. There are several challenges with live migration, including application performance, fault occurrences during VM transfer, network path selection, and security concerns. This section sheds light on how cloud computing is used to solve cloud services utilization problems and the necessity of the proposed method in detail.

### 2.1. Path Selection in Cloud Computing

Dabiah Alboaneen et al. [12] developed a solution for optimized resource utilization by integrating task scheduling and VM placements and named it Joint Task Scheduling and VMP (JTSVMP). This combined approach can achieve a better optimization solution by optimizing execution costs and resource utilization. Based on metrics such as makespan, execution cost, and degree of imbalance, the authors demonstrated the proficiency of their system. Though, this system has failed to focus on security and reliability. Irfan Mohiuddin et al. [13] presented a Virtual Machine Consolidation approach for lowering power usage by converting idle physical servers into hibernation mode (PC). A categorization technique was used to facilitate the load balance (LB) process during resource allocation and VM migration. The technique was named the Workload-Aware VMC Method (WAVMCM) since it targeted VMC based on workload to lower the number of physical machines (PMs), alleviate EC, and promote green computing. Compared to the GA centric method, the WAVMCM reduced the active servers by 9%. Nonetheless, this method was not tested in the real world; it was only checked using arbitrary data. Arabinda et al. [14] developed a load-balancing technique for a cloud-computing platform based on modified PSO to minimize the makespan and maximize resource utilization. The authors use the random dataset for evaluation purposes. In their view, security and reducing downtime are not their primary concerns. Rekha et al. [15] presented a method for reducing a system's makespan and throughput. It was created using a GA algorithm, with workload produced randomly and tested using the CloudSim simulator. It was compared to greedy and simple allocation methods already in use. Sayed Ashraf Mamun et al. designed an adaptive network-aware server consolidation system named NASCon [16]. This network uses two distinctive features of a Server-to-Server Wireless Data Center Network. A reduction of 37% in power consumption is possible. To maintain a minimal power consumption, the algorithm must be executed periodically. Liwei et al. [17] proposed a traffic-aware VM placement system to reduce network cost and energy consumption. Here, the current placement decision has been considered to predict the subsequent placement. This system uses the Amazon EC2 dataset. The simulation results show that nearly 35% of network costs and 10% of EC are reduced. This system was compared with the improved best-fit decreasing technique. Kanniga Devi et al. [18] developed a load monitoring and balancing strategy to minimize the number of migrations, message updates, network overhead, and migration costs. They proposed a new algorithm called V-MDS (a variant of minimum dominating set) to perform the load monitor. The proposed STLVM-LB algorithm was used to load balance with minimum network overhead, less migration time, and less migration cost than pre-existing techniques such as the DMA and Sandpiper algorithms; however, their focus was only on load balancing and thus not on secured migration. An accessibility model was developed by Kandoussi et al. [19] to evaluate security in virtualized frameworks. Finding the best feasible revival plan to minimize security risks and accessibility risks (or, at the very least, avoid them) was the goal. Several context studies contribute to security risks, including a Man-in-the-Middle attack and a Distributed Denial-of-Service attack. As VM relocation planning is used for revival, the results provide insight into accessibility and security risks. An optimal, secure VM deployment was proposed by Saxena et al. [20]. The Whale Optimization Genetic Algorithm (WOGA) was developed by integrating the

whale evolutionary algorithm with a non-dominated, sorting-based Genetic Algorithm. With less inter-communication delay, users can execute applications efficiently and securely across VMs, resulting in an energy-efficient allocation of physical resources. DC-MFA was incorporated into a unique load-balancing technique by Garima Verma [21]. By selecting offline VMs for migration, the servers are less likely to demand as many resources as possible. This was accomplished with DC-MFA, with six purposes: migration cost, CPU utilization, security, makespan, migration cost, and resource cost. Migration costs, energy consumption, reaction time, and security analysis were used to evaluate the performance of the suggested model.

### 2.2. Usage of Cuckoo Search in the Cloud-Computing Area

As described in [22], a Multi-Objective Cuckoo Search Optimization (MOCSO) method was proposed, which maximizes resource usage while minimizing cost. The MOCSO method is superior to state-of-the-art solutions in IaaS cloud computing for multi-objective resource scheduling. A hybridized optimization algorithm was proposed in [22] by combining the benefits of a Shuffled Frog Leaping Algorithm (SFLA) and a Cuckoo Search (CS) algorithm. Comparing the proposed system to HABCSS, CTS Alg Task, and Krill Herd, it shows an improved turnaround time, execution time, and throughput performance. Using an energy-efficient Cuckoo Search algorithm, the author proposed a method for placing virtual machines in cloud data centers [23]. A new cost and perturbation function have been added to the Cuckoo Search approach to improve it. Compared to existing algorithms, the proposed method achieves optimal placement with a lower power consumption and execution time. An overview of related works is provided in Table 1.

**Table 1.** Overview of related works.

| Author Name | Technique Used | Data Set Used | Performance Metrics | Disadvantages |
|---|---|---|---|---|
| Alboaneen, D. et al., (2021) [12] | Joint task scheduling and VM placement (JTSVMP) | NASA Ames iPSC/860 | Makespan, degree of imbalance, execution cost, Wilcoxon test | Failed to focus on Security and reliability |
| Mohiuddin, I, et al., (2019) [13] | Workload Aware Virtual Machine Consolidation Method | Random data | Power consumption, Number of active servers, Number of migrations | Test carried out through random data possibility of issues with real dataset |
| Arabinda Pradhan. et al., (2020) [14] | Modified PSO. | Random data | Makespan, resource utilization | Author left out to concentrate on throughput, number of migrations |
| SA Mamun, et al., (2021) [16] | Network-Aware Server Consolidation (NASCon) | Random data | Throughput, power consumption, consolidation cost | Security not focused, tested on random data |
| Lin, L., et al., (2020) [17] | Online traffic aware VM placement | Random data | SlotToPm, saving ratio | Security has not been focused. |
| Saxena, D., et al., (2021) [20] | WOGA | Amazon EC2 | Security threats, inter-communication cost, power consumption, and execution time | Virtual machines are not mapped to different server clusters. |

**Table 1.** *Cont.*

| Author Name | Technique Used | Data Set Used | Performance Metrics | Disadvantages |
|---|---|---|---|---|
| Garima Verma (2022) [21] | Dual Conditional Moth Flame Algorithm (DC-MFA) | Synthetic Data | Migration cost, total cost, energy consumption, response time, and security analysis | Tested on synthetic data. |
| Madni, S.H.H., et al., (2021) [24] | Multi-objective Cuckoo Search Optimization (MOCSO) algorithm | HPC2N, NASA Ames iPCS/860, SDSC | Makespan, utilization rate, resource scheduling cost. | Failed to focus on security and reliability |
| Durgadevi, P., et al., (2020) [22] | Shuffled Frog Leaping Algorithm (SFLA) and Cuckoo Search (CS) | Yardstick | Execution time, throughput, turnaround time, waiting time | Tested with limited dataset and not focused on security. |
| Salami, H.O., et al., (2021) [23] | Cuckoo Search (CS) | A. Scholl and R. Klein, Synthetic Data | Execution time, power consumption, fitness cost | Test with random dataset, failed to focus on security |

The above studies illustrate how an optimal network path could be determined to achieve multi-objective VM migration by improving resource utilization, reducing migration times, and ensuring a secure way for migration using the Cuckoo Search optimization technique.

## 3. Classical Cuckoo Search Optimization Algorithm

In recent years, traditional cuckoo search optimization algorithms and their variants [25,26] have proven to be effective metaheuristics. It is common for cuckoos to parasitize their brood on an obligate basis [27]. It is more common for cuckoos to lay their eggs in the nests of other birds than to build their own nests. Since the cuckoo's eggs hatch sooner than the host's eggs, they are provided more opportunities for survival. The cuckoo egg is viewed as a new solution compared to the host egg [28,29], in which the goal is to replace the worst solution with a better one (the cuckoo egg) [30]. The movement pattern is one of a bird's essential characteristics, and the cuckoo resembles the typical characteristics of Levy flights (LF) [31]. In Levy flight, short distances are frequent, and long distances are occasionally traveled, following a distribution with heavy tails and expanding search ranges to avoid local minima [32]. Levy flights [33] are based on the following principles:

1.  A single cuckoo egg is laid at a time and dumped into a randomly selected nest.
2.  Cuckoos lay their eggs in many nests per generation. The next generation will inherit the best nest with a satisfactory solution.
3.  Each generation has a constant number of host nests, and alien eggs are discovered according to host bird probability ($P_a \in [0, 1]$). The nest owner will either build a new nest or throw the cuckoo egg away once the egg is found.

## 4. System Model

Consider a computing environment is composed of $M$ servers, $S = \{s_1, s_2, \ldots, s_M\}$, $N$ virtual machines $V = \{v_1, v_2, v_3, \ldots, v_N\}$. Here, Vi is an instance of $V$. The VMs residing in a PM is allocated with virtual resources such as *CPU* cores, *RAM*, and disk storage. The allocated resources of a VM in a host f could be expressed as ($CPU_f$, $RAM_f$, $Disk_f$). Our proposed model needs to consider the inevitable factors such as resource availability, communication topology, and security. These factors, as mentioned earlier, are explained as sub-sections.

### 4.1. Resource Availability

The virtual machines and applications that are running on each PM can represent using its CPU usage (in MIPS), primary memory (in GB), and bandwidth usage, including its VM list, target VM index, and traffic size (in Mbps). These utilization parameters possess different scales and strategies. The present work follows our previous proposed approach [34] for calculating predicted VM resources to determine whether a PM is in a state of overload, normal load, or is underload conditions. The prediction model forecasts future resources based on recent current resource usage. This effective load-balancing technique is essential to cloud data centers to avoid unnecessary resource wastage by triggering unnecessary migrations. The respective final prediction equation performs forecasting using the proposed approach in [34], which is given as Equation (1).

$$\text{FCL}^{t+1} = \left( W_{\underset{h}{\leftarrow}} * \overleftarrow{O_L^{(t+1)}} + W_{\underset{h}{\rightarrow}} * \overrightarrow{O_L^{(t+1)}} \right) + b_{\text{FCL}} \tag{1}$$

where $\text{FCL}^{t+1}$ is the fully connected layer; $W_{\underset{h}{\leftarrow}}$ and $W_{\underset{h}{\rightarrow}}$ represent the weights of the output layer with forward and backward directions; and $b_{\text{FCL}}$ denotes the bias of the fully connected layer.

### 4.2. Communication Topology

We consider the PMs to be interconnected, and the arrangement of the PMs is represented as an undirected graph $UG = (S, E)$, where $S$ is the set of servers, represented as vertices, and $E$ is the set of links between the servers. The interconnections between the PMs are made using the bandwidth, which is assumed a weight.

### 4.3. Security

For the aforementioned cloud data center setup, there could be a possibility of a variety of security breaches that may occur during the live VM migration process. The possible security breaches may be threats or vulnerabilities over the applications and operating systems, host VMs, and co-resident VMs on various PMs via network connections. As a result of the comprehensive analysis of security risk, it is characterized by four diverse types, as follows:

#### 4.3.1. VM Risks (R1)

Based on the CVSS, VM vulnerabilities are assigned a base score according to their characteristics, and the risk experienced by the VM itself can be measured. In this case, vulnerable VMs are the preferred method for establishing co-resident VMs.

#### 4.3.2. Hypervisor Risks (R2)

An adversary can manipulate the environment's virtual machines (VMs) by gaining access to the hypervisors, making it easier for them to manage the virtual machines.

#### 4.3.3. Network Risks (R3)

In this case, network connectivity poses a threat to the system. A database server, for instance, might be compromised if a web server with the same host is captured by capturing a database server. There is a good chance that VM1 will perform web-based services over Host 1, and VM2 will perform database services over Host 2.

#### 4.3.4. Co-Residence Risks (R4)

There is a reason for this: the VMs are running on the same hypervisor. As shown in Figure 1, VM1 is a vulnerable VM, while VM2 is a regular/authorized user that shares CPU resources with VM1. Using side-channel attacks, an attacker can capture sensitive information that pertains to the authorized user.

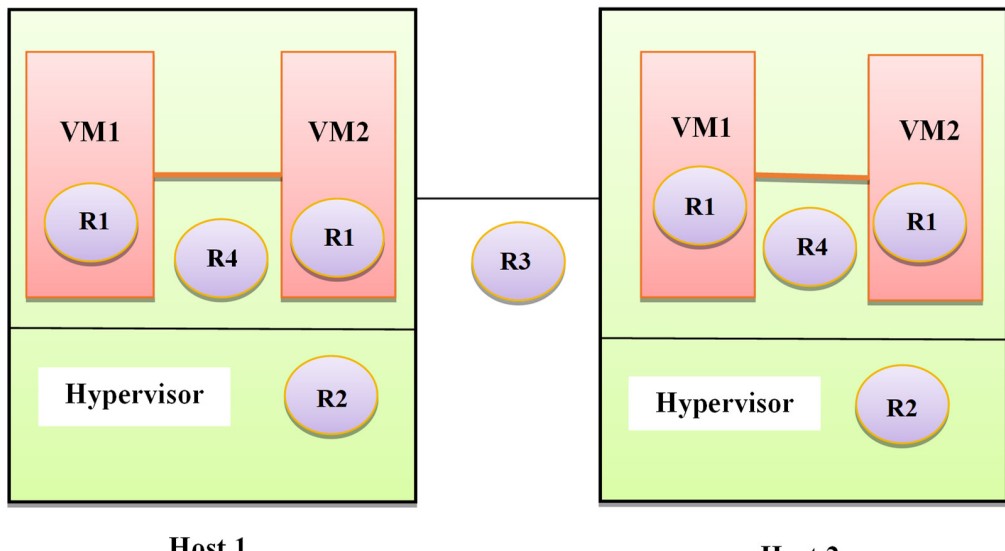

**Figure 1.** Security risk analysis model.

## 5. Proposed Network-Aware Dynamic Path Selection

Live VM migration serves uninterrupted services based on customers' requirements, in which the cloud data center problem exists in the form of optimal VM migration [2]. In general terms, a cloud data center subscriber aids in utilizing the cloud resources optimally along with uninterrupted services. Although many researchers have worked on this area, the optimal network path selection by providing authentication has not yet been explored entirely. During VM migration, the default network topology arrangement may create dense traffic among servers. There could be unnecessary migration and a waste of bandwidth due to this.

Furthermore, cloud end users will be able to feel more confident about the environment by verifying the security of the chosen network path before making a transition to the destination PM. Thus, in the current work, a new framework is proposed to provide an authenticated optimal network path for secured live VM migration. To achieve this goal, our algorithm combines a reliable host status detection algorithm with a metaheuristic algorithm to verify migration endorsements beforehand.

### 5.1. Overview of the Architecture of the Proposed Model

Figure 2 shows the overview architecture of the proposed model for a sustainable data center (SDC). In addition to checking resource availability, identifying destination PMs, and making migration decisions, migration controllers use SDC to manage cloud resources with consideration of SLAs. All three responsibilities of the migration controller possess multi-objectives, including optimal PM resource usage, less migration time, utilization of bandwidth, and network path security during live VM migration. It can be seen from Figure 2 that based on the new cloud user's request, the migration controller will perform resource availability checking. If migration is needed, the source and the respective destination server will be identified for migration execution. Then, selecting the optimal network path will be executed, followed by the authentication establishment to ensure security before migration takes place. Finally, the migration decision will be taken by the migration controller. Resource availability checking confirms the optimal resource usage among the proposed work's multi-objectives. In contrast, the proposed dynamic optimal network path-selection algorithm considers the other three objectives.

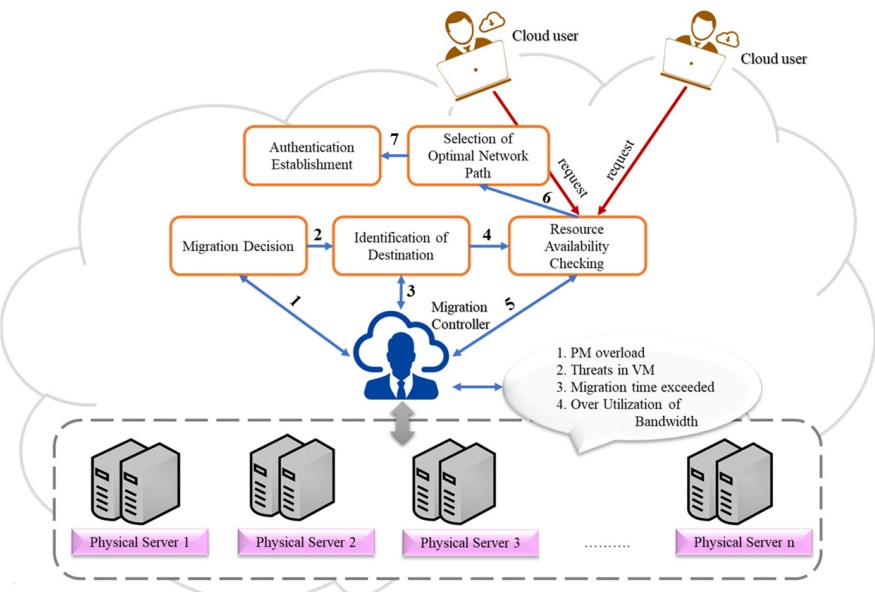

**Figure 2.** Overview architecture of the proposed system for sustainable data centers.

### 5.2. Network-Aware Dynamic Cuckoo Search Algorithm

Yang and Deb [35] proposed the multi-objective version of the Cuckoo Search (MOCS). Most engineering problems tend to be multi-objective; likewise, this research addresses finding solutions that account for multiple objectives. As the Cuckoo Search optimization algorithm is widely investigated across domains, the researchers in the cloud-computing area are also focusing on the exploration of the Cuckoo Search optimization algorithm [36,37] in the context of network path selection during LVMM.

Therefore, the proposed work extends the cuckoo search algorithm by demonstrating a novel dynamic search path to promote the goal of a suitable and secured network path selection (Figure 3). The illustration of the dynamic cuckoo search algorithm for network path-selection problems during LVMM is depicted as Pseudocode 1.

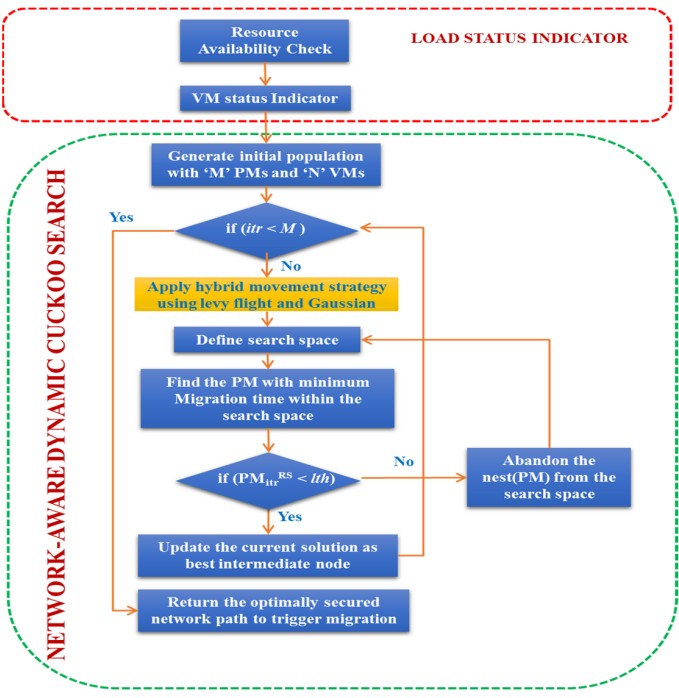

**Figure 3.** Flow diagram of the proposed Network-aware Dynamic Cuckoo Search algorithm.

---

**Pseudocode 1**: Network-aware Dynamic Cuckoo Search algorithm

---

Input:

　　　　IntN, DS, LF, GRW, RS, LTH //Intermediate Node, Destination PM, Levy Flights, Gaussian random number generation, Risk Score, Lower Threshold

---

IntN, DS, LF, GRW, RS, LTH //Intermediate Node, Destination PM, Levy Flights, Gaussian random number generation, Risk Score, Lower Threshold

---

Output:

---

Selection of optimally secured network path.

1. Begin:
2. Initializing a population of 'M' number of Physical servers, hitr (itr $\leq$ M);
3. While (itr < M) or (IntN = = DS)
4. Revise cuckoo $h_{itr}$ by LF or GRW via Equation (3);
5. SearchSpace= function Ret_SearchSpace($h_{itr}$);
6. MinMT_PM = minMigTimePM(SearchSpace)
7. Evaluate fitness ().
8. if new one is better than old
9. 　　　　　IntN = MinMT_PMitrRS.
10. 　　　　　Add IntN in the network path.
11. else:
12. 　　　　　Abandon the PM from the search space and goto step 5
13. End while
14. Return the optimally secured Network path.
15. End;
fitness (MinMT_PMitrRS)
1. Begin:
2. If MinMT_PMitrRS < LTH:
3. 　　　　　IntN = MinMT_PMitrRS.
4. Return IntN
5. End;

---

　　　　The following subsections describe the stages of the proposed Network-aware Dynamic Cuckoo Search algorithm. The stepwise flow diagram of the proposed method is shown in Figure 3.

### 5.3. Stages of the Network-Aware Dynamic Cuckoo Search Algorithm

5.3.1. Cuckoo Egg Representation

　　　　In the case of the VM allocation problem, the authors in [38] presented cloud data resources in the scenario of Cuckoo Search [9]. In the same way, with necessary changes, we adapt this representation for the dynamic path-selection problem. For this work, 'M' number of nests or physical servers have been assigned with 'N' number of virtual machines or host eggs. Each cuckoo egg represents a newly created VM with resource attributes (RAM, CPU, and disk). Each VM has an allocated migration bandwidth as one essential factor that helps during optimal network path selection. The algorithm's execution starts concerning the source and destination host servers when initialization is done. Based on the VM status indicator, the overloaded or underloaded VMs will be selected as a new egg (source VM) and the destination nest (destination pm) for migration. The generation of infeasible solutions can be avoided during VM migration by prioritizing VMs according to the available PM resources.

5.3.2. Self-Adaptive Model Parameter

　　　　The parameter setting of metaheuristics is essential in exploring and exploiting that strategy. From the classical cuckoo search algorithm, model parameters such as the number of host nests, the probability of the alien egg discovery, and the maximum iteration of the algorithm have been identified. In the case of live VM migration, the number of host nests will be treated as the number of host servers. $P_a$ is the probability of discovering a

vulnerable VM, and the maximum iteration of the algorithm is reached when the destination server encounters it. Among the three model parameters, $P_a$ can change according to the circumstances; i.e., it is a self-adaptive parameter in the proposed dynamic Cuckoo Search algorithm. Adopting a static smaller or larger $P_a$ value may increase the number of iterations, resulting in poor performance or delayed convergence. To resolve this issue, the *Pa* parameter should be fine-tuned, expanding the output vector's divergence at the last generation. Hence, the $P_a$ parameter values are calculated as self-adaptive using the number of generations [27].

$$P_a(i) = P_a max - \frac{i}{NT_a}(P_a - P_a min) \tag{2}$$

where $_{NT}$ is the number of total iterations; $_i$ is the current iteration; $P_a max$ is the maximum value of $P_a$; and $P_a min$ is the minimum value of $P_a$.

In the first step, an initial $P_a$ value for each solution is set with a constant value of 0.15, based on a previous, widely utilized path-selection study [32]. It determines a 15% probability of getting a vulnerable VM at the initial iteration. As a novelty in our proposed NDCS, the $P_a$ value will be updated by a self-adaptive strategy during every iteration in successive steps. It creates a chance of getting low or high values that can trade-off fast convergence and good performance.

### 5.4. Hybrid Movement Strategy of Cuckoo

Generally, Cuckoo Search optimization tries to identify the nest at which a cuckoo egg will hatch. Cuckoos locate nests using the conventional Levy flight distribution movement strategy. To reach the nest, it calculates the bird's step length. The novelty of the proposed algorithm rests on the cuckoo bird's movement strategy. A cuckoo bird's search area is determined by its movement strategy. A cuckoo bird searches the immediate population of nests covered under the given area. Here, we considered two movement strategies: Levy flight and the Gaussian distribution movement strategy. When exploring unknown or large-scale search spaces, Levy flights can prove to be beneficial. When the size of the population samples increases, the Gaussian distribution will be more accurate. A hybridization of Levy flight and Gaussian distribution random walk was employed to identify the destination server. By integrating the two approaches, the proposed algorithm creates ample search space instead of a single movement strategy with a dynamic step length. A dynamic step-size selection is also incorporated into the hybrid proposed technique while choosing the search space for the movement strategy.

The below equation represents the solution of the next generation hybridized strategy.

$$x_i^{(t+1)} = x_i^{(t)} + const_i \left( x_{best}^{(t)} - x_i^{(t)} \right) ss \oplus randnum(i) \tag{3}$$

where $const_i$ is a constant value and the product of the $const_i$ with the random solution, which has been obtained through the difference between $x_{best}^{(t)}$ and $x_i^{(t)}$; $ss$ is the step size; and $randnum()$ is a random number generator with a mean of 0 and a standard deviation of 1.

$$ss(i) = ss_{ma}exp(C.i) \tag{4}$$

$$C = \frac{1}{NT}ln\left(\frac{ss_{min}}{ss_{max}}\right) \tag{5}$$

The value of $_{ss}$ adopted from [10] changes dynamically based on the number of generations, increasing the output values' divergence.

Using Equation (3), a new solution will be generated, whereas a new step-size approach computes the bird's movement. In the proposed new hybrid movement strategy, the step size determines the step length of the cuckoo bird's random walk defined by two parts: one is random number generation by Gaussian distribution, and another is an arbitrary

solution difference between $x_{best}^{(t)}$ and $x_i^{(t)}$. The algorithm searches for more finely tuned paths using the redefined step length ($_{ss}$).

### 5.5. Fitness Evaluation

In our proposed NDCS, the hybrid movement strategy is applied to identify the intermediate nodes of the network path for the given destination PM. The search iteration process proceeds until the selected destination PM is reached. In general, the step length denotes the radius of the search space where the cuckoo searches for the destination nest. In the case of the proposed approach, the new step length paves a more prominent search radius using the hybrid movement strategy, to help cover a large search space area to reach the destination. As a result, the number of PMs will be covered under the selected radius, subjected to optimal node selection. The PM with a minimum migration time is considered for the upcoming fitness evaluation process. In this evaluation process, the estimated combined risk score of the chosen PM is evaluated for fitness. If the selected PM is not fit, it will result in discarding the PM as an insecure PM and recap the search for choosing the next optimal node by excluding the abandoned host. However, a successful comparison results in treating the PM as the best solution and treating it as an intermediate node until it reaches the destination.

Different attack scenarios in cloud data centers were considered for calculating the other risk scores. Attacks in cloud centers can be carried out through various attack paths, including threats or vulnerabilities in applications and operating systems, host VMs, and co-resident VMs on different PMs via network connections. The conclusion can be drawn that VM and hypervisor vulnerabilities alone cannot be considered when assessing cloud security risks.

For instance, if any intermediate PM possesses a high-risk score value, the corresponding PM will be put into an isolated zone [39]. This approach is designed to isolate the most dangerous virtual machines first and minimize the number of attack paths through a network connection. The security risk of $v_i$ is calculated as Equation (6):

$$RS^i = 1 - \left(1 - RS_1^i\right)\left(1 - RS_2^i\right)\left(1 - RS_3^i\right)\left(1 - RS_4^i\right) \tag{6}$$

The median value of all the VM risk values is considered and evaluated as Equation (7):

$$Securityrisk = median\left(RS^i\right) \tag{7}$$

where $i = 1\ to\ N$. The median value of the risk score is more robust than the mean value since the risk scores are in a skewed distribution. Next, the vulnerable VM needs to be isolated from others. Therefore, the VM with a higher risk value is susceptible.

### 5.6. NDCS Termination Criteria and Evaluation Function

While evolutionary algorithms can last forever, they also have termination criteria. A prespecified number of iterations is applied to the evolutionary algorithm to prevent premature convergence. After a predetermined number of iterations, the global best position becomes the optimal solution to the optimization problem. In this research, the termination criterion of NDCS is when it reaches the destination PM or explores all the physical hosts.

## 6. Experimental Evaluation

As part of this section, we examine the effectiveness of the proposed secured network-aware VM migration strategy. We present the dataset description, experimental setup, and experimental results.

### 6.1. Workload Dataset Description

In 2011, Google created the Google cluster trace [40], which tracks CPU and memory usage for each activity over 29 days on a cluster of 12.5k workstations. Each task is composed of one or more tasks, and hence a single job is composed of several tasks. This trace spans 29 full days and 672,075 jobs and consists of a random 1-s sampling of CPU and memory consumption taken from each task's 5-min usage reporting period.

### 6.2. Simulation Setup and Experimental Results

We have implemented the proposed technique using the CloudSim 3.03 environment. Researchers widely use this software to implement cloud environments [41]. Furthermore, the CloudSim tool provides researchers with the ability to simulate dynamic resources, network setup, and IaaS activities for methodology exploration. We experimented to see how well the proposed migration strategy works for effective network path selection to perform secured LVMM. In general, the purpose of performance metrics is to measure the performance of the proposed technique against benchmarked optimization techniques such as Cuckoo Search, variants of Cuckoo Search, Genetic Algorithm, and Particle Swarm Optimization. The migration metrics listed below were suitable for evaluating the proposed NDCS with other existing approaches.

#### 6.2.1. Resource Utilization

Resource utilization is the usage of primary resources such as the CPU, RAM, and disk.

#### 6.2.2. Energy consumption

In a cloud system, energy consumption represents the energy consumed by all ICT devices connected to the system. The energy consumption of the cloud can be calculated by personal terminals, networking nodes, and local servers.

#### 6.2.3. Makespan

Makespan is the overall time required to complete all tasks submitted to the system. It is the maximum time the host takes to run over the data.

#### 6.2.4. Total Migration Time

It includes the time it takes to migrate a VM from one resource to another and the time it takes for the VM to start executing at the destination. As more VMs are migrated, the longer the migration time will be, adversely affecting the makespan.

#### 6.2.5. Security

In live VM migration, security is a more pressing concern because it poses risks to the VM, the hypervisor, co-resident VMs, and the network. To protect a virtual machine from all these kinds of risks during the process of VM migration, the network path chosen by the migration must be secured.

In addition to migration metrics, three more factors must be analyzed to ensure the performance of the metaheuristic optimization technique proposed in this article. They are fitness value, minimum and average number of iterations, and accuracy.

### 6.3. Performance analysis of CS and Variants

Based on each resource's dimensionality and parameter fixing, the conventional Cuckoo Search and variants of Cuckoo Search have been used to evaluate the optimization and selection of network paths for live VM migration. According to the results of in-depth research on cuckoo flight variants, a comparison was made of variants of the Cuckoo Search algorithm, tabulated below (Table 2).

**Table 2.** Performance analysis chart of CS and its variants.

| Parameter | Cuckoo Search | Binary Cuckoo Search | Dynamic Cuckoo Search |
|---|---|---|---|
| Input params | $P_a$ = probability of discovery n = size of population | $P_a$ = probability of discovery n = population sigmoid function | $P_a$ = probability of discovery as dynamic $\alpha$ = step size determined dynamically. |
| Output | Improved population | Improved population | Dynamically improved population |
| Step size | Commonly used step size = 0.01 | Commonly used step size = 0.01 | Dynamically determined step size and it will be greater than 0 |
| Range | $P_a = [0, 1]$, commonly $P_a = 0.25$, $n = [10, 50]$, commonly n = 10 | $P_a = [0, 1]$, commonly $P_a = 0.25$, $n = [10, 50]$, commonly n = 10 | Range will vary based on the number of generations. $P_a\ (min) = 0.005$, $P_a\ (max) = 0.15$, $\alpha\ (min) = 0.01$, and $\alpha(max) = 0.15$ |
| Step length | In terms of step lengths, this process is governed by a power law distribution with a heavy tail. | There is a heavy tail power law distribution for the step length in this process. | As a function of the combination of the Levy and Gaussian distributions, the step length is calculated |

Since this section compares CS and its variants to prove the enhanced performance of the proposed CS algorithm over other variants, here, the convergence curve was plotted based on the fitness function of the proposed NDCS algorithm. In Figure 4, we measure the change in fitness based on the initial population size, N, as several virtual machines and the global iteration number reach the destination VM. The probability of identifying an alien egg (higher risk VM) was set as 0.15.

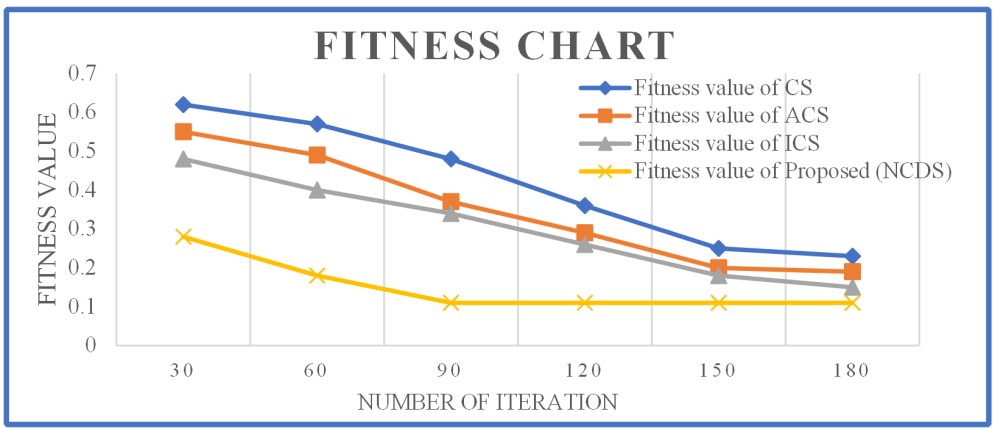

**Figure 4.** Fitness chart (fitness value vs. number of iteration).

According to Figure 4, the population fitness value tends to become stable after about 90 generations due to the fitness function, such as the total migration time and the risk scores of the proposed NDCS. Lastly, the minimum fitness value was 0.11, and the maximum objective function was 0.62. The time complexity of the model was analyzed as $O(logM)$.

### 6.4. Performance Analysis of the Migration Metrics

As the previous section has proven the outperformance of the proposed dynamic Cuckoo Search algorithm, this current section justifies applying for the proposed work on the cloud application, LVMM. This section deals with the performance of migration metrics that deal with the network path selection for LVM. When a migration controller wants to trigger migration between source and destination, the VM to be migrated should follow the secured path. To prove this objective through the proposed NDCS, the essential migration metrics, including resource utilization, power consumption, makespan, total migration time, and security, were considered for comparison with the state-of-the-art techniques.

### 6.4.1. Resource Utilization

The utilization of a sustainable data center can be described as the number of resources it consumes. When it comes to using resources efficiently, it is essential to use competent mapping by mapping the virtual machines with the pools of resources. Ultimately, it contributes to the growth of profit and revenue for cloud service providers while also meeting the needs of cloud users. In such a way, the VMs are set up to use their host's CPU, RAM, and disk to complete various tasks immediately. The sudden spike in the host's resource use led to server failure, triggering unnecessary migrations to complete the allocated task. One of the responsibilities of the migration controller admin is to maintain the usage of resources in an optimized way. Therefore, providing confidence about better resource utilization is necessary while selecting a secured network path for live VM migration. Based on the comparison, it is proven that the proposed NDCS consumes fewer resources Figure 5. Furthermore, the variation in the number of VMs of our load-forecasting mechanism using CDB-LSTM [34] provides better accuracy in forecasting the loads of the CPU, RAM, and disk. It denotes that the proposed network path selection does not affect the load-balancing work [34].

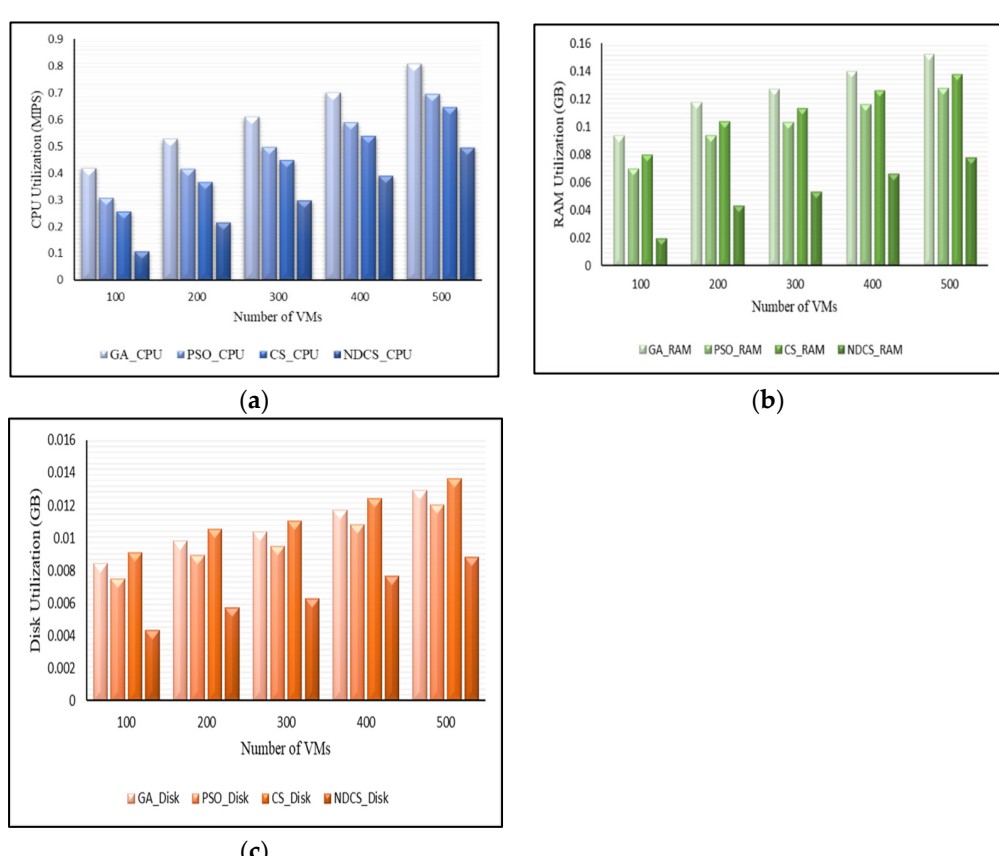

**Figure 5.** Resource utilization comparison chart: (**a**) CPU utilization vs. number of VMs; (**b**) RAM utilization vs. number of VMs; (**c**) disk utilization vs. number of VMs.

### 6.4.2. Energy Consumption

First, we investigated the total energy consumption of the simulated sustainable data center under different migration methods in [34]. Here, the purpose of checking energy consumption arises from the bandwidth utilization of each intermediate node path of the selected network path. Hence, a comparison was made by applying different optimization algorithms for network selection in Figure 6. It turns out that the all-benchmarked approaches show worse energy efficiency under every network-selection scenario. Our approach performed well in the energy comparison because, in the dynamic Cuckoo Search,

we followed the dynamic selection of the cuckoo's direction, which helps select the exact network path with the necessary intermediate nodes. Meanwhile, the dependency of the proposed fitness value on total migration time and risk score calculation omits the occurrence of bandwidth over usage (due to attackers). Thus, comparing energy consumption among the different benchmarked optimization approaches with the proposed algorithm proves that selecting an appropriate intermediate node with normal utilization reduces energy wastage during LVM.

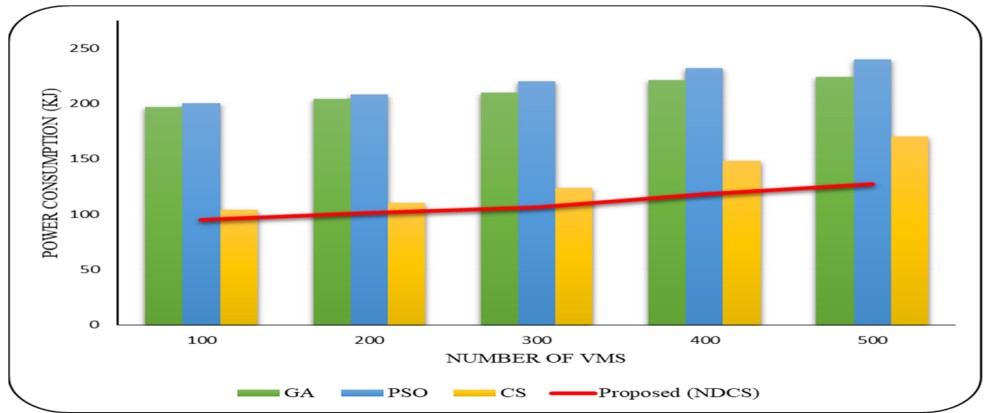

**Figure 6.** Energy utilization vs. number of VMs.

From Figure 6, the usage of dynamic flight selection helps the load-balancing approach in minimizing energy consumption to a greater extent through the varied VMs. It is evident that while the 500 VMs are in use, the proposed NDCS are consuming 127 KJ, the lowest among all the compared benchmarked approaches. Meanwhile, the traditional cuckoo is comparatively lesser than the other two approaches, such as PSO and GA.

### 6.4.3. Makespan

Figure 7 compares the makespan time of the proposed NDCS against existing approaches. The simulation was conducted using a varying number of virtual machines, ranging from 100 to 500. The existing three optimization algorithms, GA, PSO, and CS, were used for comparison with the proposed NDCS. As the number of virtual machines increases, the algorithm's makespan time rises, indicating that the VMs have a shorter makespan time. The unit of measure of makespan time is seconds. When virtual machines increase in number, the proposed NDCS algorithm generates a lower makespan compared only to the other techniques that were considered.

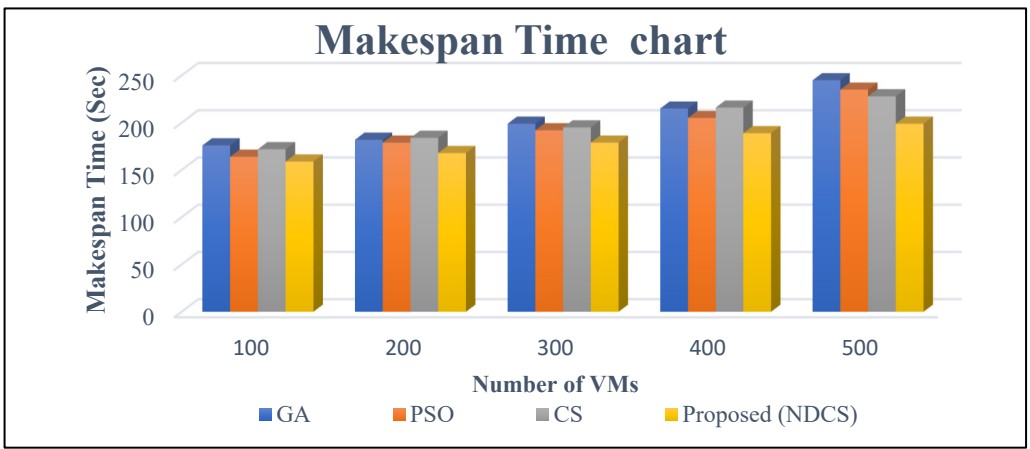

**Figure 7.** Makespan vs. number of VMs.

Figure 7 expresses that the makespan time of the proposed NDCS is 199 s when the number of VMs is 500, which is less compared to the other state-of-the-art techniques. The basic cuckoo and PSO also possess the lowest makespan time than GA due to the fast-converging nature of these algorithms. Due to these results, the proposed NDCS enables cloud users to save more money in the cloud-computing environment.

### 6.4.4. Total Migration Time

During the migration process, the total migration time is the time taken from the point the migration starts to the time the virtual machine resumes at the destination, and when the source can be discarded. According to Figure 8, for different optimization algorithms, the total migration time is recorded for varying virtual machines using cloud simulation on Google cluster data. Regarding the total migration time, random migration has the worst performance across data centers [34], but it shows the best performance among heuristic approaches and the proposed work.

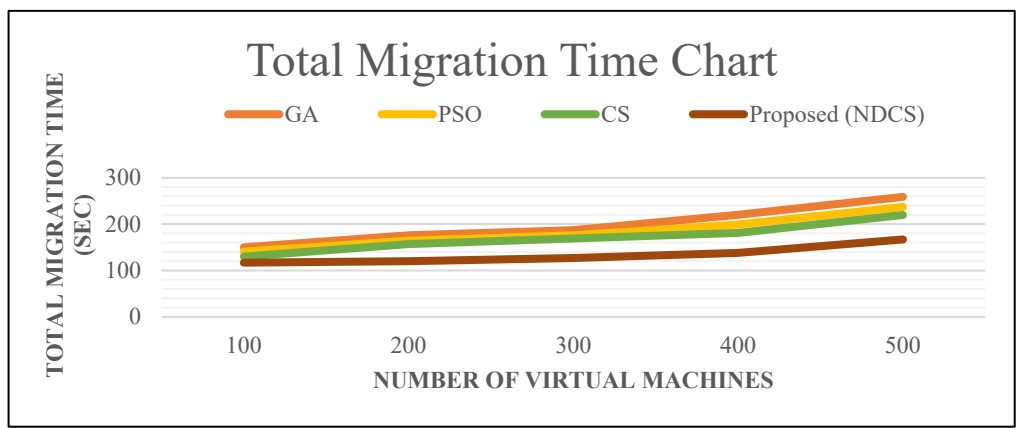

**Figure 8.** Total migration time vs. number of VMs.

From Figure 8, we can infer that the proposed NDCS achieves a lower total migration time of 167 s compared to other techniques. The factor behind the lower total migration time is that the proper load was forecasted and the destination selected for migration in our prior work, and the optimized network path was identified from the proposed metaheuristic algorithm NDCS. Thus, we can conclude the NDCS network-selection algorithm can be implemented as it outperforms others when choosing an intermediate node, leading to a lower total migration time.

### 6.4.5. Security Analysis

Figure 9 illustrates the impact of using a dynamic Cuckoo Search algorithm for path selection in terms of ensuring security during LVMM. The performance is measured through simulation of the factors with a median risk score and diverse paths. Ideally, if the median risk score of the group is 1, then the highest possibility of a risk increase exists. To ensure the path's security, we determined that four combinations of sources and destinations be considered. An assessment was conducted to assess the significance of the proposed NDCS over other state-of-the-art techniques, and the assessment results were recorded.

Figure 9 expresses the four different sources and destinations that were used. Our proposed NDCS has lower risk scores in all source and destination scenarios compared to the different benchmark approaches such as Genetic Algorithm, Particle Swarm Optimization, and the classic Cuckoo Search algorithm.

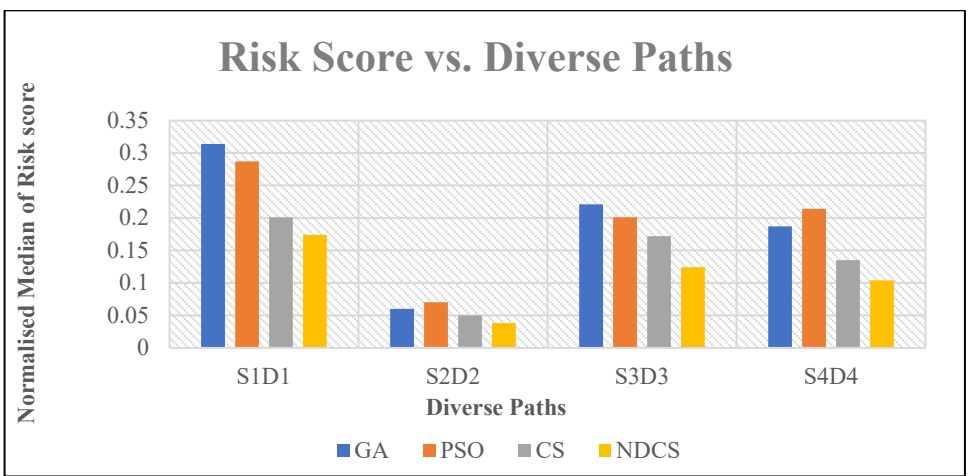

**Figure 9.** Security metric (risk scores vs. diverse paths).

## 7. Conclusions

The proposed Network-aware Dynamic Cuckoo Search (NDCS) optimization algorithm may allow cloud-computing platforms to identify a secure network path by reducing the energy consumption, maximizing resources, decreasing the total migration time, and minimizing the duration of the makespan. The principal contribution is a hybrid movement strategy, which employs both Levy light and Gaussian strategies, improving the capability to search and speed up convergence. In addition to that, in the proposed system (NDCS), four risk scores, such as VM risk, hypervisor risk, network risk, and co-residence risks, have been identified. A combined risk score was calculated and utilized as a fitness criterion to assess how well the proposed system will ensure migration security. The entire process was assessed using a simulation system known as CloudSim. The experiments were carried out using Google cluster data to verify the efficiency of the proposed system. From the results obtained, it is noticeable that when compared to the other state-of-the-art techniques, the proposed model has less power consumption, averaging 127 kJ, lesser use of resources, a makespan duration of 199 s, and a total of 167 s of migration time. The proposed model also has considerably lower risk scores than the benchmark algorithms, such as Genetic Algorithm (GA), Particle Swarm Optimization (PSO), and the classic Cuckoo Search algorithm. Future work could improve this research by using different SLA measures and compressing methods to enhance this LVMM procedure.

**Author Contributions:** Conceptualization, N.V.S. and V.S.S.S.; methodology, N.V.S.; software, N.V.S.; validation, N.V.S. and V.S.S.S.; formal analysis, N.V.S.; investigation, N.V.S.; resources, N.V.S.; data curation, N.V.S.; writing—original draft preparation, N.V.S.; writing—review and editing, V.S.S.S.; visualization, N.V.S.; supervision, V.S.S.S.; project administration, V.S.S.S.; funding, N.V.S. All authors have read and agreed to the published version of the manuscript.

**Funding:** This research received no external funding.

**Conflicts of Interest:** The authors declare no conflict of interest.

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
