# Peer review of "An Effective Secured Dynamic Network-Aware Multi-Objective Cuckoo Search Optimization for Live VM Migration in Sustainable Data Centers"

_sustainability, doi:10.3390/su142013670_

Round 1

Reviewer 1 Report

In this work, the authors have proposed a new framework to provide an authenticated optimal network path for secured live VM migration. Overall, this research topic is relevant, and the work proposed is of good quality. Here are a few suggestions for improving the paper's readability:

There are a few English grammar errors or hard-to-follow sentences that need to be corrected.

Explain all the acronyms at the first occurrence. E.g., what is PM?

The paper's contribution is not completely clear. Based on the bullet points in Section 1, it appears that this work has only one contribution.

I'm not sure what Table 1 is for. It might be useful if the authors also include relevant works from section 2.2.

Figure 2 is difficult to comprehend. Arrows must be labeled with the correct order number.

The figure references must be corrected; they are incorrect.

It is difficult to determine what security measures are considered when making an LVMM decision.

Author Response

In this work, the authors have proposed a new framework to provide an authenticated optimal network path for secured live VM migration. Overall, this research topic is relevant, and the work proposed is of good quality. Here are a few suggestions for improving the paper's readability:

Comment 1: There are a few English grammar errors or hard-to-follow sentences that need to be corrected.

Response 1:

               Thank you for your valuable suggestion. We have done the rectification of grammatical errors all over the manuscript.

Comment 2:  Explain all the acronyms at the first occurrence. E.g., what is PM?

Response 2:

As per your suggestion, we have incorporated the explanation for all acronyms

Comment 3: The paper's contribution is not completely clear. Based on the bullet points in Section 1, it appears that this work has only one contribution.

Response 3:

               Contributions to this manuscript are,

  • Finding the optimal network path for VM migration
  • Proposed dynamic cuckoo search approach that combines levy flights and gaussian distribution to form an innovative search strategy.
  • The risk score factor is assessed to select intermediate nodes in establishing the optimal secured path.
  • Ensuring a secure network path for live VM migration with less energy consumption and optimal resource utilization
  • Various objectives apart from optimal network path and security other factors such as energy consumption, and optimal resource utilization also accomplished.

Comment 4:  I'm not sure what Table 1 is for. It might be useful if the authors also include relevant works from section 2.2.

Response 4:

               As per your valuable suggestion, other references are also included in table 1.

Comment 5: Figure 2 is difficult to comprehend. Arrows must be labelled with the correct order number.

Response 5:

               As per your comments, arrows are labelled with order number in figure 2.

Comment 6:

               The figure references must be corrected; they are incorrect.

 Response 6:

               Thanks for your suggestion. The figure references are corrected.

Comment 7:

It is difficult to determine what security measures are considered when making an LVMM decision.

Response 7:

Thank you for your valuable comment, we have adopted the combined risk score for each PM as fitness criteria to ensure secure LVMM

Reviewer 2 Report

you must change the following references...

1, 27, 29, 30, 31, 32, 33, 38, 40, because they are older than 5 year

the conclusions defined are not very complete, some additional factor should be added to differentiate it from the papers of the cuckoo algorithm used for its work.

Author Response

Comment 1: You must change the following references: 1, 27, 29, 30, 31, 32, 33, 38, 40, because they are older than 5 years.

Response 1:

               Thank you for your valuable suggestion. We have changed all the references which are older than five years in the manuscript.

Comment 2: The conclusions defined are not very complete, some additional factor should be added to differentiate it from the papers of the cuckoo algorithm used for its work.

Response 2:

As per your suggestion, we have modified the conclusion part and included the same in the revised manuscript.

Reviewer 3 Report

To enhance the effectiveness and security of live virtual machine migration, this paper utilizes a network-aware meta-heuristic dynamic multi-objective cuckoo search algorithm to select the optimal dynamic network path for security. Compared with existing strategies, the developed hybrid movement strategy enhances the search ability by expanding the search space and adopting the combined risk score estimate of each PM as a fitness criterion to ensure safety and fast convergence. Experimental results show that the proposed method outperforms existing methods in terms of energy consumption, migration time, and resource utilization.

There are still some problems in this paper that need to be improved:

1) This article considers four types of security risks. It is suggested to explain the reasons for choosing these types of security risks, and analyze whether there are other types of security risks.

2) It is suggested to analyze the time complexity of the algorithm described in Figure 3.

3) The titles of Section 5.2 and Section 5.3 are the same, is it a typo?

Author Response

To enhance the effectiveness and security of live virtual machine migration, this paper utilizes a network-aware meta-heuristic dynamic multi-objective cuckoo search algorithm to select the optimal dynamic network path for security. Compared with existing strategies, the developed hybrid movement strategy enhances the search ability by expanding the search space and adopting the combined risk score estimate of each PM as a fitness criterion to ensure safety and fast convergence. Experimental results show that the proposed method outperforms existing methods in terms of energy consumption, migration time, and resource utilization.

There are still some problems in this paper that need to be improved:

Comment 1: This article considers four types of security risks. It is suggested to explain the reasons for choosing these types of security risks and analyze whether there are other types of security risks.

Response 1:

               Thank you for your valuable observation. Attackers can compromise a cloud VM in a variety of ways. They can exploit vulnerabilities in the OS or applications carried by a VM, co-residence VMs, a host VMM, or VMs on multiple physical machines connected to a cloud VM, so we have considered four types (VM Risk, Hypervisor Risk, Network Risk, and Co-Residence Risks) of security risks.

Comment 2:  It is suggested to analyse the time complexity of the algorithm described in Figure 3.

Response 2:

Thanks for your suggestion, the time complexity of the algorithm is O(log M) and it is included in section 6.3 of the revised manuscript

Comment 3: The titles of Section 5.2 and Section 5.3 are the same?

Response 3:

               Thanks for your observation, it’s a typo error and the proper title has been included in the revised manuscript.

Round 2

Reviewer 1 Report

Thanks for replying to my previous comments. Still I see a couple of issues that need to be fixed:

1. Use same style of writing e.g., use either data center or data centre not both.

2. Figures are still referenced incorrectly. E.g., there is no Figure 9.

Author Response

The authors thank the reviewers for their valuable observations to enhance the quality of the paper. Hereby we are submitting the point-wise response describing the changes incorporated based on the reviewer comments.

Comment – 1

Use same style of writing e.g., use either data center or data centre not both.

Response 1:

               The authors have changed all the occurrences with the same style as “data center”, which has been highlighted in the manuscript.

Comment -2

Figures are still referenced incorrectly. E.g., there is no Figure 9.

Response 2:

               Thanks for your observation, the figure numbers are modified and cited properly in the revised manuscript
